# OpenReview forum: "The Personality Illusion: Revealing Dissociation Between Self-Reports & Behavior in LLMs"
_ICML.cc/2026/Conference — ICML 2026 regular_

### Official Review · Reviewer_1jjw · 2026-03-05

**Soundness:** 3
**Presentation:** 4
**Significance:** 3
**Originality:** 3
**Overall Recommendation:** 4
**Confidence:** 5

**Summary:**

This paper investigates whether the "personality" traits that LLMs claim to possess in self-report questionnaires are actually reflected in their objective behavior. While previous research often relied on these self-reports at face value, the authors conduct a systematic evaluation of 18 models, including both base (pre-training) and instruction-aligned (post-training) versions, to test the predictive validity of these traits.

The study finds a significant "dissociation" between self-reported persona and actual action. The authors demonstrate that while instructional alignment (such as RLHF and instruction tuning) systematically shifts a model's self-reported personality (e.g., increasing "Openness" and "Agreeableness"), these scores fail to reliably predict how the model performs in behavioral tasks. Even when the researchers used "persona injection" to nudge a model toward a specific trait, the resulting behavioral changes were often inconsistent with the intended identity. Ultimately, the paper argues that LLM personality is an "illusion"—a byproduct of alignment that creates linguistic plausibility without grounding it in stable, functional behavior.

### Summary of Contributions

- Empirical Characterization of Trait Evolution: The authors trace how human-like traits emerge and stabilize between pretraining and post-training stages, finding that instruction-tuning reduces run-to-run variability compared to base models.


- Identification of the "Personality Illusion": The study formalizes the gap between self-report and behavior, showing that only about 24% of trait-task associations are statistically significant, and nearly half of those diverge from human patterns.


- Evaluation of Persona Controllability: The paper quantifies the effect of persona injection, proving that while it successfully steers linguistic self-reports, it has a minimal or inconsistent impact on real behaviors like sycophancy or risk-taking.

**Compliance With Llm Reviewing Policy:**

Affirmed.

**Final Justification:**

I have raised my rating to a Weak Accept following the rebuttal. I specifically appreciated the additional experiments on OLMo’s intermediate checkpoints (Base $\rightarrow$ SFT $\rightarrow$ DPO $\rightarrow$ Instruct), which were much more convincing than the results in the original manuscript. The paper is exceptionally well-written and the authors clearly put a lot of systematic effort into the evaluation. This clarity, combined with their robust experimental setup, makes the findings very accessible and easy to follow.

However, I still have reservations regarding the paper's timeliness and the choice of the Big Five framework. Having followed this area for years, I note that several works had already explored these directions prior to the submission deadline. Furthermore, I find the Big Five personality traits to be a bit too straightforward and coarse-grained to reveal truly deep or insightful conclusions about the mental traits of LLMs. Nevertheless, the systematic nature of the study and the clarity of its conclusions make it a worthwhile contribution to the field.

**Key Questions For Authors:**

Please refer to my weaknesses section

**Limitations:**

Yes (in the appendix)

**Strengths And Weaknesses:**

Weaknesses

- The paper lacks timeliness as its three core research questions appear to have been extensively explored in recent literature. Specifically, several studies from 2024 and 2025 have already investigated personality emergence across training stages, the predictive validity of self-reports, and the impact of persona interventions.

E.g.,  for the RQ1:
 Uncovering the Computational Ingredients of Human-Like Representations in LLMs (https://arxiv.org/pdf/2510.01030) Specifically identifies instruction-finetuning as the most crucial ingredient for "human-model alignment" in semantic representations compared to architecture or multi-modal training. (pay attention to Figure 2a, Figure 3 in this paper)

"The Persona Selection Model: Why AI Assistants might Behave like Humans" (Anthropic), Proposes the Persona Selection Model (PSM): Pre-training teaches the LLM a distribution over many possible personas (latent), while post-training (alignment) "selects" and stabilizes a specific "Assistant" persona.

https://aclanthology.org/2024.tacl-1.43.pdf Investigates how instruction tuning (IT) and RLHF introduce or amplify human-like cognitive biases (e.g., decoy effect, belief bias) that were absent or weaker in base models.

Human Traits in Artificial Minds: Personality Construction in Contemporary LLMS (https://sciety.org/articles/activity/10.21203/rs.3.rs-8210799/v1) offers a systematic review concluding that while base models have broad/unstable linguistic patterns, instruction fine-tuning is the stage that constructs a "consistent, predominantly prosocial, and emotionally stable" profile.

Similarly, for the RQ2:

"Predictive Validity of LLM Personality Reports: A Case Study on Classic Economic Games" directly tests if self-reports predict behavior in strategic games (e.g., Prisoner’s Dilemma). Finds that self-reports capture "surface-level textual regularities" but are poor predictors of actual cooperation or risk-taking compared to "observer reports."


"Large Language Models Display Human-Like Social Desirability Biases in Big Five Surveys" discovers that LLMs "fudge" their self-reports to appear more socially desirable (e.g., higher Agreeableness). This bias makes self-reports an unreliable proxy for actual behavioral tendencies.


"Evaluating the Validity of Self-report Scales in Measuring Personality Design in LLM-based Chatbots" shows weak correlations between a chatbot's self-reported scores and human-perceived personality during real interactions, questioning the criterion validity of these scales for AI.

Similarly, there are a lot of work discussing injecting persona to (system) prompt to instruct LLM's behaviors (RQ3).

- The paper mainly use Big-5 and self-regulation for personality trait study. However, I think those dimensions are too coarse-grained and they can not realistic guided ai agent's behaviors. I suggest to include more detailed traits such as value, morality, satefy, etc, such as the OpenAI's paper "persona features control emergent misalignment"

- In the study of comparing pretrained model and post-trained models seem questionable. pretrained models are mainly for knowledge learning, their instruction-following ability is not well. So a lower performance may just drive from their poor instruction-following aspect, not their intrinsic personality traits. A suggested improvement is for all the pretrained base model, you can applied a small scale of instruction-tuning stage which excluding any human preference related data (e.g, you can use pure math dataset)

-----------------

Strengths


- Exceptional Clarity and Presentation: The manuscript is remarkably well-written, structured, and easy to follow. The authors succeed in making complex psychometric and statistical concepts accessible to a machine learning audience. Furthermore, the visual presentation is a significant asset; the figures are high-quality, aesthetically pleasing, and effectively distill large-scale experimental results into intuitive, clear insights.

- Comprehensive and Systematic Evaluation: The scale of the study is impressive. By evaluating 18 different LLMs across various families (LLaMA, Qwen, Mistral, etc.) and sizes, the authors provide a robust experiment that captures the current state of the field.

- Clear and Actionable Insights: The paper provides a formal, quantitative foundation for the "Personality Illusion." While the gap between self-report and behavior has been discussed qualitatively in the community, this work provides specific metrics that move the discussion toward a more empirical and grounded understanding of AI "traits."

---

> ### Author Rebuttal · Authors · 2026-03-30
>
> We thank Reviewer 1jjw for the detailed review and for highlighting the exceptional clarity, systematic evaluation, and actionable insights of our work. We are happy to address the concerns below.
>
> 1. _The three RQs have been extensively explored in recent literature_
>
> **Response:** We thank the reviewer for the survey. We note that the majority of cited works are **concurrent or post-submission**: "Persona Selection Model" (Anthropic, Feb 2026), "Predictive Validity of LLM Personality Reports" (ANLP, Mar 2026), "Human Traits in Artificial Minds" (Nov 2025), and "Human-like Representation" (Oct 2025) were all published after or around our submission deadline; **per ICML policy, comparison with concurrent work is not expected.** For the 2024 works: the TACL paper focuses on cognitive decision biases rather than personality, and Salecha et al. and Zou et al. are already cited and discussed in our paper. **Regardless of time, all these works study fundamentally different constructs** — none examine the systematic relationship between self-reported personality traits and cross-domain behavioral outcomes central to our work. As the reviewer recognized, it is precisely this "comprehensive and systematic" integration that reveals the "personality illusion" and moves the discussion from qualitative intuition to empirical grounding.
>
> **Action:** We will discuss recent developments in the revision and are **happy to discuss specific cited works in more detail if helpful.**
>
> 2. _Big Five and self-regulation are too coarse-grained; finer-grained traits should be included._
>
> **Response:** We appreciate this question and are glad to provide clarifications. We want to note that the dissociation we test operates at the level of broad personality traits, not fine-grained behavioral instructions. Role/behavior-based instructions (e.g., "you are a cautious person who avoid gambling") can steer targeted behavior, but personality traits (e.g., "you are open") represent **cross-situational tendencies** — a person high in openness should be more curious, risk-taking, and receptive to new ideas across diverse contexts. Big Five is designed to capture exactly this cross-domain consistency. As instructions become more fine-grained, they naturally become more effective — but that tests instruction-following, not personality. It would be interesting to explore where this boundary lies, but that is a complementary direction.
>
> Measuring and steering broad personality traits is an active area in LLM research, with the Big Five framework being widely adopted in the CS/AI community [1,2,3,4]. Our goal was to test its predictive power, which we found to be limited. We’ve extended beyond Big Five by including Self-Regulation, HEXCAO and discuss broader frameworks like Schwartz’s value model as future directions (Appendix B).
>
> **Action:** We will strengthen the discussion of scope and future directions in the revision.
>
> 3. _Comparing pretrained and post-trained models is questionable as it can be due to the base model’s poor instruction following._
>
> **Response:** First, for RQ1, **we don’t claim to identify the specific mechanism by which post-training shapes traits**,  but confirm that models express more consistent and coherent personality trait profiles in self-reports. This motivates RQ2: can these consistent self-reports predict behavior? Pinpointing exactly what instruction-tuning vs. RLHF each contributes is an important but separate open question.
>
> Second, **base models in our experiments successfully follow questionnaire instructions**. We adapted the prompt format for base models (Table 9, Appendix I), and they reliably output single numbers from 1–5. The trait variability we observe is not a failure to comprehend the task.
>
> To further address this, we run an **additional experiment** to analyze OLMo's intermediate checkpoints (base → SFT → DPO → Instruct). We observe a gradual progression: variance steadily decreases (e.g., agreeableness SD: 0.40 → 0.33 → 0.15 → 0.13), while traits shift toward higher agreeableness (3.30 → 3.93 → 4.18 → 4.25) and lower neuroticism (2.87 → 2.43 → 2.45 → 2.39). This pattern is consistent across all six traits and confirms that trait stabilization is not simply an artifact of improved instruction-following, but reflects cumulative effects across post-training stages.
>
> **Action:** We will include the OLMo analysis and clarify the claim of RQ1 in the main text.
>
> [1] Personality Traits in Large Language Models
>
> [2] PersonaLLM: Investigating the Ability of Large Language Models to Express Personality Traits
>
> [3] Evaluating and Inducing Personality in Pre-trained Language Models
>
> [4] Big5-Chat: Shaping LLM Personalities Through Training on Human-Grounded Data
>
> We hope these clarifications and new experiments address the reviewer's concerns. We are happy to answer any further questions. **If you find our responses convincing, we would be grateful if you would consider updating your score accordingly.**

---

> > ### Author Rebuttal · Reviewer_1jjw · 2026-04-02
> >
> > Thank you for the detailed response. I appreciate the additional experiments analyzing OLMo’s intermediate checkpoints (Base → SFT → DPO → Instruct), as these results are more convincing than those currently displayed in the paper. It is clear that the authors have put a great deal of effort into conducting this systematic research;  I don't have a strong opinion to reject the paper so I have raised my rating to Weak Accept.
> >
> > However, my initial two concerns regarding timeliness and the use of the Big Five personality test remain unchanged. Having followed this field for years, I am aware that even pretty prior to the ICML submission deadline, there were already several public research works exploring this same direction. Additionally, I hope that in the future, more nuanced frameworks than the Big Five will be used to study mental traits in LLMs, as I believe that would lead to more meaningful results.

---

> > > ### Author Response · Authors · 2026-04-06
> > >
> > > We thank the reviewer for acknowledging our clarifications and for raising the score! We also appreciate the recognition of the systematic effort behind this work and the positive reception of the OLMo checkpoint experiments. We want to take this opportunity to further address the remaining concerns.
> > >
> > > **On timeliness.** We maintain that the concurrent works cited were published at or after our submission deadline and thus fall outside ICML's prior-art requirement. We will nonetheless strengthen the related work to situate our contribution more precisely, emphasizing the systematic, cross-domain, quantitative aspects that distinguish our empirical grounding of the personality illusion from single-game or qualitative studies.
> > >
> > > **On Big5 coarseness.** We directly address this by extending RQ2 (self-reports predict downstream behavior) to four additional instruments, tested on two large instruction-tuned models (Claude 3.7 Sonnet and Qwen-2.5-72B-Instruct). The new instruments:
> > > - **HEXACO**: an alternative personality questionnaire
> > > - **Moral Foundations Questionnaire (MFQ)**: a well-established instrument for morality, addressing the reviewer's suggestion on moral traits
> > > - **Schwartz's Basic Human Values (PVQ)**: a well-established framework for human values, addressing the reviewer's suggestion on value-based traits
> > > - **Theory of Planned Behavior (TPB)**: a fine-grained behavioral intention instrument measuring attitude, perceived behavioral control, subjective norm, and intention toward a concrete behavior
> > >
> > > **Broad trait and value frameworks (HEXACO, MFQ, PVQ).** Across all three, self-reports produce coherent, instrument-appropriate profiles but **largely fail to predict CCT behavior**.
> > > - **HEXACO**: Models report moderate Emotionality (M=4.76/7) and Agreeableness (M=4.14/7), with low within-model variability (avg. SD=0.18 for Claude, 0.05 for Qwen). After model controls, **no trait predicts CCT risk-taking** (ΔR²=+0.018; all p≥.113).
> > > - **MFQ**: Models report high Care (M=5.42/6) and Fairness (M=5.57/6), with lower Authority (M=3.00/6) and Purity (M=2.34/6). After model controls, **no moral foundation predicts CCT behavior** (ΔR²=+0.006; all p≥.340).
> > > - **PVQ**: Models report high Universalism (M=5.70/6) and Benevolence (M=5.61/6), with low Power (M=2.19/6). After model controls, most values are non-significant (ΔR²=+0.068); Achievement (β=−0.49, p=.028) and Security (β=+0.63, p=.014) survive, treated as exploratory given n=40. Still, **the 10 values together only explain 6.8% of the variance**.
> > >
> > > **TPB (behavior-proximal instrument).** Models report moderate Attitude (M=4.76/7) and high PBC (M=5.19/7), with low Subjective Norm (M=2.72/7). After model controls (ΔR²=+0.031), PBC (β=+0.36, p=.002) and SN (β=−0.24, p=.016) survive, while Attitude and Intention do not. This is interpretable: models reporting higher confidence in executing the specific policy do flip more cards within-model. We view this as a **constructive boundary finding** — fine-grained, behavior-specific instruments may capture action-proximal self-knowledge better than broad trait questionnaires. **However, such instruments are limited as they require knowing the behavior context in advance, unlike the broad traits we study.**
> > >
> > > We emphasize that these extensions are **exploratory and preliminary**, but they suggest an interesting pattern: broad psychological frameworks still often fail to predict behavior, while behavior-proximal instruments (TPB) suggest some potential, though effects remain very modest. This could be an interesting near-future research direction. However, deeply extending these across all tasks, models, and RQs requires careful design beyond the scope of a rebuttal and is itself a distinct research question worth studying. We will include these exploratory results in the camera-ready version to motivate future work.
> > >
> > > We selected Big5 in this paper because it is widely used by many influential works to evaluate or steer LLM personality. We complemented it with Self-Regulation, whose well-established relations to Big Five in human psychology allowed us to ground our findings in human benchmarks.
> > >
> > > In conclusion, we believe our **additional explorations following the reviewer's suggestions provide valuable initial patterns and insights for future research.** We hope these demonstrate our commitment to addressing the concerns thoroughly, and we would be grateful if the reviewer could consider these new results in their final assessment. Thank you very much again for the constructive feedback that directly motivated these extensions.

---

### Official Review · Reviewer_1sRZ · 2026-03-10

**Soundness:** 4
**Presentation:** 4
**Significance:** 3
**Originality:** 3
**Overall Recommendation:** 5
**Confidence:** 4

**Summary:**

This study systematically examines whether large language models (LLMs) exhibit coherent personality traits and whether such traits correspond to behavioral consistency. Using the Big Five personality framework and self-regulation measures, the research investigates three dimensions: the origin, manifestation, and control of LLM personality. Results show that instruction alignment substantially stabilizes self-reported personality profiles, increasing openness, agreeableness, and self-regulation while reducing neuroticism, and decreasing run-to-run variance by up to 90%. However, behavioral experiments—including risk-taking, social bias, honesty, and sycophancy tasks—reveal weak and inconsistent relationships between reported traits and actual behavior. Furthermore, persona injection effectively shifts questionnaire responses but produces minimal or non-significant behavioral changes. Overall, the findings indicate that LLMs can generate linguistically coherent personality profiles, yet these profiles do not reliably predict decision-making behavior, suggesting that apparent LLM “personality” largely reflects a surface-level linguistic phenomenon rather than a stable behavioral construct.

**Compliance With Llm Reviewing Policy:**

Affirmed.

**Final Justification:**

I thank the authors for their thorough and constructive rebuttal, which has significantly strengthened the paper. Overall, the study is methodologically sound and ambitious, combining psychometric assessments with behavioral experiments to examine the nature of LLM “personality.” My initial concerns regarding construct validity and experimental robustness have been largely addressed. In particular, the clarified validation framework and use of persona injection as a control provide convincing evidence that the observed dissociation between self-reported traits and behavior is unlikely to be a measurement artifact, but rather reflects a genuine phenomenon.

In terms of originality and significance, the paper makes a compelling contribution by systematically demonstrating that linguistically coherent personality profiles do not reliably translate into consistent behavioral patterns. The tripartite framework (origin, manifestation, control) is well-motivated, and the finding of structural dissociation offers an important perspective on the limitations of current alignment methods. The rebuttal further enhances the work by introducing the idea that alignment may induce a latent psychometric structure, while also framing persona–behavior asymmetry as a useful diagnostic signal for evaluating model robustness and safety.

The rebuttal directly addressed my main concerns and positively changed my evaluation. The authors not only strengthened the empirical rigor but also deepened the theoretical interpretation of their findings. While limitations remain—particularly regarding weak behavioral correlations—they are appropriately acknowledged. Overall, I find this work to be clear, rigorous, and high influence, and I support its acceptance.

**Key Questions For Authors:**

1.	Latent Psychometric Structure: Does the finding that instruction alignment significantly stabilizes the personality traits of large language models suggest that alignment procedures implicitly embed a human-like latent structure within the linguistic representation space of model?
2.	Structural Dissociation: Might the discrepancy between persona-based self-reports and actual behavioral performance be better explained by measurement artifacts or task design limitations, rather than by a structural decoupling between the generative linguistic system and the causal decision-making processes of large language models?
3.	Persona Injection: Can the observed asymmetry between persona-conditioned self-reports and relatively invariant behavioral outputs provide a useful diagnostic framework for evaluating the robustness and safety of AI systems?

**Limitations:**

The research investigates the "personality illusion" in Large Language Models (LLMs) by exploring whether their stable linguistic expressions represent genuine behavioral dispositions or merely surface-level pattern matching. The experimental framework reveals that while instructional alignment significantly stabilizes self-reported trait profiles and fosters inter-trait correlations that mirror human psychological patterns, these developments primarily consolidate linguistic consistency rather than behavioral depth. Despite the apparent coherence of these trait profiles, a fundamental dissociation exists between self-reported traits and actual performance in downstream behavioral tasks—such as risk-taking, social bias, and honesty—where alignment with human expectations frequently hovers near chance levels. Furthermore, although targeted interventions like persona injection successfully steer self-reported identities, they exert minimal or inconsistent influence on action-oriented decision-making, confirming that surface-level trait manipulation does not translate into functional reliability. Ultimately, the study underscores that current alignment methodologies, such as RLHF, refine linguistic plausibility without grounding it in behavioral regularity, thereby necessitating a shift toward more behaviorally anchored alignment strategies to ensure functional consistency in real-world deployments.

**Strengths And Weaknesses:**

Strengths:
• Reduced Variance: Alignment significantly cuts down on random fluctuations in how the model describes itself.
• Methodological Ambition: The study doesn't just rely on surveys; it adapts classic psychological tasks to test if the AI's "behavior" matches its "words."
• Post-Alignment Shift: Evidence suggests that personality-like traits aren't just accidental leftovers from pre-training; they are actively reshaped and induced by the alignment process.
• Practical Relevance: The important application for anthropomorphizing LLMs and highlighting the caution required when interpreting linguistic coherence as indicative of genuine psychological structure.

Weaknesses:
• Construct Validity: Reliance on questionnaire-based self-reports raises questions about construct validity; it is unclear whether outputs reflect genuine latent model properties or simply learned human-like response patterns
• Experimental Robustness: Behavioral results show weak, inconsistent, and often statistically insignificant relationships between reported traits and actual decision-making, suggesting that the methodology may not fully capture meaningful personality-behavior connection.
• Measurement Limitations: Localized correlations between traits and behavior lack robustness across tasks and architectures, highlighting potential limitations in the experimental design or measurement sensitivity.

---

> ### Author Rebuttal · Authors · 2026-03-30
>
> We thank Reviewer 1sRZ for their thorough engagement. We are encouraged by the recognition of our methodological ambition and the acknowledgment of the practical relevance of our findings to LLM anthropomorphization. We believe the remaining concerns are primarily a matter of clarification and targeted additional analysis, addressed in detail below.
>
> 1. _Do self-reports reflect genuine latent model properties or superficial human-like response patterns?_
>
> **Response:** This is the core question of our study. While our core investigation is on self-report and behavior consistency as a metric of how genuine the latent properties are, we also validate self-reports themselves: (1) LLMs respond across a **meaningful range without floor/ceiling compression** (Figure 8), and (2) Big5 constructs show theoretically expected inter-correlations with Self-Regulation —conscientiousness positive, neuroticism negative — consistent with prior work [1] (Fig 2 C, Sec 2.3c). These **surface-level checks establish linguistic consistency.** However, RQ2 and RQ3 are specifically designed to test the deeper claim — whether self-reports reflect latent behavioral properties — and show they do not. We hypothesize, consistent with the reviewer's framing, that this linguistic consistency reflects **sophisticated pattern-matching rather than genuine internalized traits**, as discussed in Sec 5 ("Illusions of Coherence through Alignment and Prompting").
>
> 2. _Are the weak trait-behavior associations a genuine null result or a methodological artifact?_
>
> **Response:** We took five steps to ensure the null results **reflect genuine dissociation rather than design limitations.**
>
> (1) **Self-report validity:** RQ1 replicates known alignment-induced trait shifts consistent with prior work, confirming the self-report methodology is sound.
>
> (2) **Construct coherence:** Big5 traits correlate with Self-Regulation in human-expected directions (Fig. 2C; Sec 2.3c), confirming inter-construct consistency.
>
> (3) **Task engagement:** LLMs produce diverse, non-compressed behavioral responses across all tasks — no floor/ceiling effects (visually Fig 8, p.24), with distinct model profiles visible in Figures 6–7. We also provide **full distributional statistics**, including floor/ceiling frequencies and between-model discrimination tests, in our response to **Reviewer WQAG**.
>
> (4) **Controlled perturbation grid:** Repeated probing across matched prompts, temperatures, and seeds ensures results are not noise-driven (Sec 2.1).
>
> (5) **Positive control via RQ3:** Persona injection produces large, significant self-report effects within the same experimental setup, demonstrating the framework can detect large effects when they exist — the null behavioral results are therefore not a power failure.
>
> Together, these checks establish that the weak trait-behavior associations **reflect genuine dissociation rather than methodological limitations.**
>
> 3. _Can the methodology be used for AI robustness and safety testing?_
>
> **Response:** We agree this is a promising direction and appreciate the reviewer framing it constructively. The asymmetry we observe — persona injection reliably shifts self-reports but not behavior — can indeed serve as a diagnostic: it reveals the gap between what a model claims to be and how it actually acts, **which is directly relevant to alignment evaluation.** A model reports low risk-taking or high agreeableness under a persona prompt but behaves identically to its default condition is, in a meaningful sense, misrepresenting itself. This has safety implications for persona-based deployments where users may calibrate trust based on a model's stated identity. We see this as an important avenue for future work.
>
> **Action:** Will add a brief note to the Discussion section highlighting the diagnostic potential of the dissociation framework for alignment and safety evaluation.
>
> 4. _Does post-training alignment implicitly embed a human-like latent personality structure in LLMs?_
>
> **Response:** This is a **compelling and plausible interpretation.** Our findings in RQ1 show that alignment does not merely shift trait means — it stabilizes trait expression, reduces run-to-run variance by 81–90%, and strengthens inter-trait correlations in directions consistent with human psychology (Fig. 2C). This pattern is consistent with alignment implicitly organizing the model's linguistic representation space around something resembling a coherent personality structure, rather than producing isolated trait shifts. However, we are cautious about concluding that this constitutes a genuinely latent structure in the representational sense, especially since our work shows a clear dissociation between surface-level self-reports and actual task behavior.
>
> **Action:** We will add a brief note in the Discussion acknowledging this interpretation and pointing to representation-level investigation as an important direction for future work.
>
> [1] The Power of Personality.

---

> > ### Author Rebuttal · Reviewer_1sRZ · 2026-04-03
> >
> > I would like to thank the authors for their rigorous and comprehensive rebuttal. Your response substantially address my concerns regarding construct validity and methodological robustness, thereby strengthening the manuscript’s empirical foundation. Regarding to latent psychometric structure, I appreciate the authors’ willingness to incorporate the hypothesis that alignment procedures may implicitly organize the model’s linguistic representational space. Although the observed behavioral dissociation warrants cautious interpretation, recognizing this “latent structure” as a potential byproduct of alignment—and identifying representation-level analysis as a critical direction for future research—adds meaningful theoretical depth to the Discussion. In terms of structural dissociation, the authors present compelling evidence that the observed dissociation is not attributable to measurement artifacts. In particular, the five-step validation framework—including replication of established alignment effects and the use of persona injection as a positive control—demonstrates a high level of methodological rigor. Collectively, these results support the interpretation of “structural decoupling” as a genuine phenomenon within the examined architectures. In considering the diagnostic relevance for AI safety, the authors' conceptualization of the asymmetry between expressed personal and functional behavior as a form of model misrepresentation is particularly commendable, as it offers a rigorous lens for evaluating alignment consistency and system robustness in real-world deployments. Their contribution beyond descriptive psychological characterization toward a practically applicable assessment framework. Overall, the authors demonstrate strong scientific rigor and responsiveness, since they substantially strengthen the manuscript’s contribution to the literature by delineating the theoretical boundaries of “LLM personality” and substantiating their claims with robust statistical evidence.

---

> > > ### Author Response · Authors · 2026-04-06
> > >
> > > We are deeply grateful for the reviewer's generous and detailed acknowledgement. We are particularly encouraged by the recognition of our "five-step validation framework" and the framing of the self-report/behavior asymmetry as "model misrepresentation" relevant to AI safety — this resonates strongly with the direction we hope this work will inspire. We will incorporate all discussed additions, including the latent structure discussion and the diagnostic framework for alignment evaluation, in the camera-ready version. Thank you for the engagement that genuinely helped sharpen our contribution.

---

### Official Review · Reviewer_WQAG · 2026-03-13

**Soundness:** 3
**Presentation:** 3
**Significance:** 4
**Originality:** 4
**Overall Recommendation:** 5
**Confidence:** 4

**Summary:**

This paper systematically investigates whether LLMs possess meaningful "personality" by examining three research questions: (RQ1) how trait-like profiles emerge across training stages, (RQ2) whether self-reported traits predict behavior on psychologically-grounded tasks, and (RQ3) whether persona injection can steer both self-reports and behavior. The study evaluates 18 models (6 base, 6 small instruct, 6 large instruct) using Big Five and Self-Regulation questionnaires alongside behavioral tasks (risk-taking, social bias, honesty, sycophancy). The key finding is a "personality illusion": instructional alignment stabilizes self-reported traits and makes them more human-like, but these traits fail to predict behavior (~24% significant associations, with only 52% of those matching human directions), and persona injection shifts self-reports but not behavior.

**Compliance With Llm Reviewing Policy:**

Affirmed.

**Final Justification:**

I believe the authors’ responses have addressed my concerns, and I consider this work to have meaningful contributions to the field.

**Key Questions For Authors:**

1. How sensitive are your RQ2 results to the choice of human-expected directions? Some trait-task mappings have mixed evidence even in human psychology (marked as "?" in Table 7). If you exclude uncertain mappings and focus only on well-established associations (e.g., neuroticism to reduced self-regulation), does the alignment rate change meaningfully?

2. Could the behavioral tasks themselves have floor/ceiling effects that limit trait expression? For example, if most models show very low sycophancy or very high risk-aversion due to alignment training, there may be insufficient variance to detect trait associations. Reporting the distribution of behavioral scores per task would help assess this.

3. Does the dissociation hold for within-model variation or only between-model variation?

**Limitations:**

No specific suggestions could be added, given the revealing nature of this paper.

**Strengths And Weaknesses:**

Strengths:
-  The paper addresses a novel question: whether LLM "personality" as measured by self-report questionnaires has any behavioral validity. The framing as a dissociation between linguistic self-expression and behavioral consistency is precise and actionable.
-  The three-RQ structure is conceived. RQ1 establishes that traits exist and stabilize; RQ2 tests whether they matter; RQ3 tests whether they can be controlled. The use of 18 models spanning base/instruct and small/large, three prompting conditions, three temperatures, and three runs per condition  demonstrates attention to robustness.
- The behavioral tasks are adaptations of established psychological paradigms with known trait-behavior mappings from human research. The paper provides extensive literature grounding for expected directions of trait-task associations, enabling principled evaluation rather than post-hoc interpretation. This interdisciplinary integration is a clear strength.

Weaknesses:
- The RQ1 findings are somewhat expected. That RLHF/instruction tuning increases agreeableness, reduces neuroticism, and stabilizes outputs is consistent with the known objectives of alignment training (producing helpful, harmless, honest responses). The finding that aligned models show stronger trait intercorrelations is more novel, but the overall RQ1 contribution is confirmatory.
- The behavioral tasks may not be well-calibrated for LLMs. While the psychological grounding is strong, several tasks may not translate cleanly to text-based LLM evaluation.
- The paper assumes that if LLMs had "real" personality, the same trait-behavior associations observed in humans should hold. But human trait-behavior links arise from motivational, affective, and embodied processes that LLMs lack entirely. Even if an LLM had internally consistent behavioral tendencies, there is no reason these should follow human-specific mappings.

---

> ### Author Rebuttal · Authors · 2026-03-30
>
> We would like to thank Reviewer WQAG for appreciating our framing as “precise and actionable”, praising our “attention to robustness”, and recognizing that “interdisciplinary integration is a clear strength.” We believe the remaining concerns are mostly a matter of clarification and added analysis. We provided additional details for each concern raised.
>
> 1. _RQ1 findings are expected and confirmatory._
>
> **Response:** We agree there is a confirmatory aspect to RQ1, which was intentional. We wanted to **establish the validity of our methodology** and align with existing work. Nevertheless, there is also a novel part, as the reviewer noted, in testing the theoretically expected, but yet untested, inter-correlation between separate psychometric questionnaires - Big 5 traits and Self-Regulation (SRQ).
>
> Action: We will rephrase the results in RQ1 to better emphasize the confirmatory goal and the novel part.
>
> 2. _Do behavioral tasks translate to LLM evaluation, and could floor/ceiling effects limit variance?_
>
> **Response:** We address this on two levels. First, each task has established LLM precedent: risk-taking via the CCT [1]; honesty via confidence calibration [2]; sycophancy via Asch-style moral dilemmas [3]; and IAT-based stereotyping [4]. Second, LLMs actively engage with all tasks. In Fig 8 (p. 24); all tasks use 35–86% of their theoretical scale range with <1% of observations at floor/ceiling, and Kruskal-Wallis tests confirm significant between-model differences (all p < .001, η² = 0.83–0.91, ICC = 0.79–0.97). The dissociation cannot be attributed to insufficient variance.
>
> **Action:** We will add full distributional statistics as a new table in Appx H.4 (see below).
>
> | Task | Scale | Mean (SD) | % range used | % at floor | % at ceiling | ICC | KW η² | Sig |
> |---|---|---|---|---|---|---|---|---|
> | Risk Taking | 0–32 | 21.9 (5.5) | 71% | 0.0% | 0.0% | 0.79 | 0.83 | *** |
> | Stereotyping | −1–1 | 0.2 (0.4) | 81% | 0.3% | 0.0% | 0.93 | 0.87 | *** |
> | Sycophancy | 0–100 | 48.9 (19.6) | 86% | 0.6% | 0.0% | 0.91 | 0.86 | *** |
> | Epistemic Honesty | −100–100 | 33.9 (20.4) | 35% | 0.0% | 0.0% | 0.97 | 0.91 | *** |
> | Self-Reflective Honesty | 0–100 | 21.5 (13.0) | 60% | 0.9% | 0.0% | 0.87 | 0.86 | *** |
>
> 3. _Is the dissociation genuine, or do LLMs simply have non-human trait-behavior mappings?_
>
> **Response:** We **do not claim LLMs should follow human patterns;** we ask whether associations established in human literature actually hold in LLMs, given the widespread practice of applying human psychometric instruments to LLMs and the implicit assumption that persona steering transfers to behavior. Our findings challenge this assumption, contributing to two questions: (1) **how valid are human-derived alignment expectations** for LLMs, and (2) to what extent **do persona injections produce genuine behavioral consistency.** Whether LLMs have their own internally consistent trait-behavior tendencies that diverge from human mappings is an exciting direction we discuss in Sec 5 and Appx C.
>
> 4. _Are RQ2 results robust to excluding uncertain trait-task mappings?_
>
> **Response:** We **graded all 30 trait-task associations** from Appx L into three evidence tiers: **Strong** (meta-analytic support, 6 pairs), **Medium** (multiple consistent individual studies, 19 pairs), and **Weak/unclear** (single study, mixed or qualified findings, 5 pairs). Restricting to Strong-evidence pairs yields 54.5% alignment (95% CI: 35–73%), and to Strong+Medium yields 52.9% (44–62%) — both indistinguishable from the full-sample alignment rate reported in the paper and overlapping with the 50% chance baseline. The **results are not sensitive to the inclusion of uncertain mappings.**
>
> **Action:** Evidence grading rules, the full grading table, and sensitivity analysis will be added to Appx H.4.
>
> 5. _Does the dissociation hold within models, not just between models?_
>
> **Response:** The RQ2 analysis (Sec 3.4) is **already a within-model test.** Both levels are visually inspectable in Fig 8 (p. 24) - large dots (per-model means), small dots (individual runs). To **formally test between-model dissociation**, we run a Mundlak decomposition separating within- and between-model trait components via mixed-effects models, GEE, and OLS with cluster-robust SEs: zero significant between-model associations (0/30 pairs, all three methods), despite between-model correlations being larger in magnitude (mean |r| = 0.22, up to r = 0.34). ICC analysis confirms 77% of trait and 89% of task variance is structured between models, yet produces no systematic trait–behavior association. **The dissociation holds at both levels.**
>
> **Action:** We will add clarifying sentences to Sec 3.4 and 3.5, and the full decomposition to the new Appx H.4
>
> [1] Exploring Variability in Risk-Taking with Large Language Models
>
> [2] Alignment for Honesty
>
> [3] Social Sycophancy: A Broader Understanding of LLM Sycophancy
>
> [4] Measuring Implicit Bias in Explicitly Unbiased Large Language Models

---

> > ### Author Rebuttal · Reviewer_WQAG · 2026-04-03
> >
> > I believe the authors’ responses have addressed my concerns, and I consider this work to have meaningful contributions to the field.

---

> > > ### Author Response · Authors · 2026-04-06
> > >
> > > We sincerely thank the reviewer for the kind acknowledgement and for recognizing this work as having "meaningful contributions to the field." We are very encouraged by this and will make sure the camera-ready version reflects all the improvements discussed, including the distributional statistics, evidence grading sensitivity analysis, and the Mundlak decomposition. Thank you again for the thoughtful questions that helped us strengthen the paper.

---

### Official Review · Reviewer_eie6 · 2026-03-13

**Soundness:** 2
**Presentation:** 3
**Significance:** 3
**Originality:** 3
**Overall Recommendation:** 4
**Confidence:** 3

**Summary:**

This paper studies whether personality-like traits expressed by LLMs through self-reports are meaningfully reflected in downstream behavior. The paper examines this question from three perspectives: how trait profiles emerge across training stages, whether self-reported traits predict behavior in real-world-inspired tasks, and whether persona injection can control both self-reports and behavior. The central claim is that alignment and prompting can stabilize and steer surface-level self-reports, but these effects transfer only weakly and inconsistently to actual behavior, suggesting a limited correspondence between linguistic self-presentation and behavioral consistency.

**Compliance With Llm Reviewing Policy:**

Affirmed.

**Final Justification:**

The rebuttal resolved most of my concerns. Accordingly, I have updated my final score.

**Key Questions For Authors:**

- The main claim is that self-reported personality in LLMs transfers only weakly to downstream behavior. How do the authors distinguish this conclusion from a possible limitation of the behavioral tasks themselves?

- The paper adopts personality constructs and survey frameworks from human psychology. Could the authors clarify more explicitly what is being transferred from humans to LLMs at the conceptual level, and what validity limits should be kept in mind when interpreting these constructs in machine systems?

- The persona injection results show that self-reports can be shifted more easily than behavior. Under what conditions do the authors believe persona prompting meaningfully changes downstream behavior, and when does it remain only a surface-level linguistic effect? A more fine-grained analysis would strengthen the work's practical implications.

**Limitations:**

Yes.

**Strengths And Weaknesses:**

Strengths

1. The paper addresses a timely and important question. The gap between self-reported personality and actual behavior in LLMs is highly relevant to alignment, evaluation, and anthropomorphism.

2. The paper is well organized. The three research questions provide a clear structure, and the overall narrative is easy to follow from motivation to conclusion.

3. Rather than relying only on questionnaire-style self-reports, the paper includes psychologically grounded behavioral tasks, including risk-taking, stereotyping, honesty-related probes, and sycophancy.

4. The main empirical message is interesting and practically important: self-reports can look coherent and human-like, yet remain weak predictors of behavior. This is a meaningful caution for current LLM evaluation practices.

Weaknesses

1. The main conclusion is intuitively appealing, but the current evidence does not fully isolate the source of the observed dissociation. In particular, it remains unclear how much of the gap comes from limitations of the behavioral tasks themselves rather than from a genuine lack of stable trait-behavior correspondence in LLMs.

2. The mapping from human personality theory to LLM behavioral tasks is still somewhat indirect. While the paper is careful and well-motivated, the validity of transferring these psychological constructs to machine systems remains debatable and warrants stronger discussion in the main text.

3. Although the paper covers several tasks, the empirical scope remains limited relative to the breadth of the paper’s framing. The paper argues for a broad “personality illusion,” but the evidence is concentrated on a specific set of survey instruments and behavioral probes.

---

> ### Author Rebuttal · Authors · 2026-03-30
>
> We sincerely thank Reviewer eie6 for the thoughtful and constructive review. We are encouraged by the recognition of our work as addressing a "timely and important question" with an "interesting and practically important" empirical message. We are very happy to address reviewer eie6's questions below.
>
> 1. _The dissociation may stem from task limitations_
>
> **Response:** We thank the reviewer for this important question. We believe the evidence supports a genuine dissociation rather than a task artifact.
>
> (1) **Behavioral scores span wide ranges** with no floor/ceiling compression (35–86% of scale used; <1% at extremes; Fig. 7–8, App. D). Occasional significant effects (Table 7) further confirm **meaningful task engagement and the tasks have sufficient discriminative power.** Full statistics are provided in our response to Reviewer WQAG.
>
> (2) Weak trait–behavior associations are consistent across all five tasks spanning distinct domains; task-specific issues would yield sporadic, not systematic, patterns.
>
> (3) Tasks are adapted from well-validated human paradigms testing widely different behavioral aspects and have been widely used in prior LLM studies.
>
> **Action:** We will move distributional analyses from Appendix D to the main text and strengthen discussion of task coverage and future extensions.
>
> 2. _Transferring human constructs to machines can be invalid_
>
> **Response:** We agree and this is precisely our point. **(1)** We do not assume LLMs should behave like humans. Rather, we empirically test an assumption already widespread in practice [1,2,3] (transferring human questionnaires to assess LLM personality) and find it could be problematic. **(2)** Additionally, the dissociation we found holds independently of human comparisons: LLMs' self-reported traits show minimal association with their own behavior regardless of whether we reference human patterns. Human comparisons serve a diagnostic purpose, identifying where and how LLM behavior might diverge, which is especially important given deployment in sensitive contexts (mental health, education).
>
> **Action:** We will make sure to clarify this point further in our introduction and strengthen this discussion in the revision.
>
> 3. _The empirical scope is limited to several instruments and when persona injection meaningfully changes behavior vs. remaining surface-level._
>
> **Response:** We appreciate this suggestion and are happy to elaborate.
>
> First, as discussed in Q1, our behavioral tasks span distinct and practically important domains for LLM deployment: decision-making (CCT), implicit social bias (IAT), epistemic calibration, metacognition, and social conformity (sycophancy). LLMs engage meaningfully with all tasks, and the observed dissociation arises consistently across these diverse domains, indicating that **the effect is robust rather than driven by any single task or narrow setting.**
>
> Second, we chose Big Five because it is by far one of the most widely adopted frameworks for studying LLM personality in the AI/CS community [1,2,3,4]. We note that our study already goes beyond Big Five by including the Self-Regulation Questionnaire (SRQ).
>
> Following the reviewer's suggestion, we conducted an _**additional experiment**_ using HEXACO on LLaMA-3-8B base and Instruct: alignment reduces variance (SD 0.23→0.11) and shifts traits toward prosocial profiles (e.g., Honesty-Humility 3.03→3.49, Agreeableness 2.40→2.76), yet traits still fail to predict behavior (no significant effects for IAT or CCT). **This confirms the dissociation generalizes across instruments.**
>
> Third, we persona prompting meaningfully changes behavior when it functions as a direct, domain-specific instruction (e.g., "you are a cautious person who avoids gambling" → reduces risk-taking on CCT). However, it remains surface-level when the expected behavioral change requires cross-situational generalization — e.g., when a "cautious" persona should also lead to higher conformity or lower overconfidence.
>
> **Action:** We will include the HEXACO analysis in the revision and strengthen the discussion of conditions under which persona effects remain surface-level.
>
> [1] Personality Traits in Large Language Models
>
> [2] PersonaLLM: Investigating the Ability of Large Language Models to Express Personality Traits
>
> [3] Evaluating and Inducing Personality in Pre-trained Language Models
>
>
> We hope these clarifications and new experiments address the reviewer's concerns. We are happy to answer any further questions. **If you find our responses convincing, we would be grateful if you would consider updating your score accordingly.**

---

### Decision · Program_Chairs · 2026-04-30

**Decision:**

Accept (regular)

**Comment:**

This paper presents a systematic study on whether LLM “personality” is behaviorally meaningful, and introduces the notion of a “personality illusion,” i.e., the dissociation between self-reported traits and actual behavior. Reviewers consistently agree that the paper is clearly written, methodologically strong, and provides a large-scale and careful empirical evaluation across models and tasks.

While some concerns were raised about novelty and the use of coarse-grained personality frameworks, these are relatively minor compared to the overall contribution. In particular, several reviewers highlight that the paper goes beyond prior work by providing a comprehensive and quantitative characterization of the phenomenon, as well as its implications for alignment and anthropomorphism.

The authors’ rebuttal addressed the main technical concerns and further strengthened the empirical evidence. Overall, the reviewers are broadly supportive, and the paper is considered a solid and valuable contribution.